# Learn Together, Stop Apart: a Novel Approach to Ensemble Pruning

## Abstract

Gradient boosting is the most popular method of constructing ensembles that allow getting state-of-the-art results on many tasks. One of the critical parameters affecting the quality of the learned model is the number of models in the ensemble, or the number of boosting iterations. Unfortunately, the problem of selecting the optimal number of models still remains open and understudied. In this paper, we propose a new look at the hyperparameter selection problem in ensemble models. In contrast to the classical approaches that select the universal size of the ensemble from a hold-out validation subsample, our algorithm uses the hypothesis of heterogeneity of the sample space to adaptively set the required number of steps in one common ensemble for different regions of data points individually. Experiments on popular implementations of gradient boosting show that the proposed method does not affect the complexity of learning algorithms and significantly increases quality on most standard benchmarks up to 2%.

## 1 Introduction

There are still many areas where classical machine learning algorithms prevail over deep neural networks despite the dramatic growth of their usage in artificial intelligence research. One of such classical algorithms is Gradient Boosting (GB) (Friedman (2001)). It allows to obtain high-quality models on table data with no multimedia (e.g., images, audios, videos), with samples full of categorical features, noisy features and labels, missing data (Zhang & Haghani, 2015; Li et al., 2007; Babajide Mustapha & Saeed, 2016). Also, the undoubted advantage of the boosting method is the low computational complexity of training and inference (Deng et al., 2018). For these reasons, Gradient Boosting is widely used in ranking (Chapelle & Chang, 2011), recommender systems (Cheng et al., 2014), meta-learning (LeDell & Poirier, 2020), and many other tasks (Touzani et al., 2018; Trofimov et al., 2012; Ling et al., 2017).

In recent years, many hyperparameters and additional options have been proposed for GB influencing the performance of the learned model (Ke et al., 2017; Ibragimov & Gusev, 2019). But the learning rate (weight of each model in the ensemble) and the size of the ensemble are the key ones. Large models are responsible for revealing complex dependencies in the data but require more time for training and inference (Friedman, 2002). In comparison, the smaller ones are less expressive but more time-efficient. The standard approach to select an optimal number of training steps is to control the quality of the model by measuring it on a hold-out sample called validation set, which is separate from the training data. The idea is to set a large enough size of the model and find the moment (overfitting point) when the validation score stops growing and begins going down. Then one can prune the ensemble to the retrieved number of iterations.

The described method has a significant and surprisingly understudied weakness. The approach assumes the existence of a universal ensemble size equally effective for any instance in the sample. In other words, the hypothesis is that all samples require approximately the same number of learners to fit them well. However, in practice, the learning task can consist of different subtasks, which correspond to different regions in the input space of the dataset, where examples follow different distributions with diversified complexities and functional dependencies. In particular, the data space may contain regions of both simple and complex surfaces for training. For the first ones, the ensemble needs a relatively small number of boosting rounds to be trained well, while the latter requires a way longer path until convergence. In this case, the generic boosting size selected by the least regret

principle is a compromise between simple and complex areas. This approach encourages models with a composition of overfitted and underfitted regions in the dataset.

To handle this issue, we propose a new method to prune large GB models based on an adaptive choice of the optimal size of the ensemble. As in the standard version of GB (Friedman, 2001), we train one sequence of learners in an ensemble but apply a different number of learned models to different regions in the dataset. Namely, we build an additional model that divides the input space into regions where the distribution of data points has homogeneous complexity and representativity. Then we optimize the ensemble size to each region individually. Our method incurs meager computational costs and can be easily incorporated into any existing learning pipeline. We apply the proposed approach to state-of-the-art open-source GB algorithms and demonstrate its ability to outperform on popular publicly available benchmarks consistently. We show that the described problem of the universal stopping moment highly affects the quality of trained models. To the best of our knowledge, this is the first research devoted to adaptive, instance–wise early stopping in GB, and we hope this paper will encourage further research of the GB algorithm.

The rest of the paper is organized as follows. Section 2 introduces notations and background on GB. Previous works on early stopping and ensemble pruning are discussed in Section 3. In Section 4, we reveal the details of the proposed approach and present theoretical reasoning and discussions. In Section 5, the effectiveness of the algorithm is empirically studied using several popular datasets. Section 6 makes conclusions and proposes ideas for future work.

## 2 BACKGROUND

In this section, we introduce necessary notations and briefly discuss basic concepts concerning gradient boosting and cross-validation for clarity and independent reading reasons.

### 2.1 GRADIENT BOOSTING

Let $\mathcal{S} = \{\boldsymbol{x}_i, y_i\}_{i=1}^n$ be a sample from some fixed but unknown distribution $P(\boldsymbol{x}, y)$, where $\boldsymbol{x}_i = (x_i^1, ..., x_i^m) \in \mathbb{X}$ is an $m$-dimensional feature representation and $y_i \in \mathbb{Y}$ is a target value of the $i$-th observation. The classical formulation of the learning problem consists in constructing a function $F : \mathbb{X} \to \mathbb{Y}$ minimizing the expected target prediction error, which is calculated using a loss function $L : \mathbb{Y} \times \mathbb{Y} \to \mathbb{R}_+$:

$$\mathcal{L}(P, F) := \mathbb{E}_{(\boldsymbol{x}, y) \sim P}[L(F(\boldsymbol{x}), y)] \to \min_F$$

Since the distribution $P$ is not given and the sample $S$ is the only source of data, the task reduces to *empirical risk minimization problem*:

$$\hat{\mathcal{L}}(\mathcal{S}, F) = \hat{\mathbb{E}}_{(\boldsymbol{x}, y) \sim \mathcal{S}}[L(F(\boldsymbol{x}), y)] = \frac{1}{n} \sum_{i=1}^n L(F(\boldsymbol{x}_i), y_i) \to \min_F$$

The ability to achieve smaller value of an empirical risk is bounded by the complexity of the set $\mathcal{F}$ from which the desired function $F \in \mathcal{F}$ is selected. A common approach to increase the expressiveness of the learned model is to build a composition (or an *ensemble*) of functions from $\mathcal{F}$. *Gradient Boosting (GB)* constructs an ensemble $F_B$ of size $B$ as a weighted sum of base functions $\{f_1, f_2, ..., f_B\} \subset \mathcal{F}$:

$$F_B(\boldsymbol{x}) = \sum_{i=1}^B \alpha_i f_i(\boldsymbol{x}) \tag{1}$$

When the set of available base functions $\mathcal{F}$ is closed under scalar multiplication, multipliers $\alpha_i$ are usually fixed and equal: $\forall i \; \alpha_i = \alpha$, where $\alpha$ is a hyperparameter of the GB algorithm called *learning rate*. Having constructed the first $t - 1$ terms, the learning algorithm aimes to select the next function $f_t$ sequentially as a solution of:

$$\hat{\mathcal{L}}(\mathcal{S}, F_{t-1} + f_t) = \frac{1}{n} \sum_{i=1}^{n} [L(F_{t-1}(\boldsymbol{x}_i) + f_t(\boldsymbol{x}_i), y_i)] \to \min_{f_t}$$

The approximate solution of the latter equation in GB is usually constructed as follows. Algorithm calculates first and second order derivatives of $\hat{\mathcal{L}}$ at the point $F_{t-1}$ w.r.t. predicted values $\hat{y}$: $g_i^t = \frac{\partial L(\hat{y}_i, y_i)}{\partial \hat{y}_i}\Big|_{\hat{y}_i = F_{t-1}(\boldsymbol{x}_i)}, h_i^t = \frac{\partial^2 L(\hat{y}_i, y_i)}{\partial \hat{y}_i^2}\Big|_{\hat{y}_i = F_{t-1}(\boldsymbol{x}_i)}$, and selects a least squares estimator to Newton's gradient step in the functional space:

$$f_t = \arg\min_{f \in \mathcal{F}} \sum_{i=1}^{N} h_i^t(\vec{x}_i, y_i) \left( f(\vec{x}_i) - \left( -\frac{g_i^t(\vec{x}_i, y_i)}{h_i^t(\vec{x}_i, y_i)} \right) \right)^2, \tag{2}$$

see (Chen & Guestrin, 2016) for details.

## 2.2 Model selection via cross-validation

Since the quality estimation based on a train set used in the learning process is biased (Prokhorenkova et al., 2017) (the inference is performed on the unseen data), it is conventional to use a separate independent set, called *validation set*, to control the generalization ability of the algorithm. The whole dataset $\mathcal{S}$ is split into two disjoint sets $\mathcal{S}_{train}$ and $\mathcal{S}_{valid}$, where the first one is used for learning and the latter for quality estimation.

The final result of this procedure is often highly dependent on the particular train-validation split and, therefore, quality estimation can be very noisy. To tackle this issue, one can use *cross-validation* (Stone, 1974) method: split the data $\mathcal{S}$ into $k$ subsets of approximately equal size, or *folds*, $(\mathcal{S}_1, \mathcal{S}_2, ..., \mathcal{S}_k)$ s.t. $\mathcal{S} = \bigsqcup_{i=1}^{k} \mathcal{S}_i$, and perform $k$ rounds of training-evaluation cycle using $\mathcal{S}_{-i} := \mathcal{S}/\mathcal{S}_i$ as the training set and $\mathcal{S}_i$ as the validation data for each $i \in \{1, 2, ..., k\}$. Then the estimated quality is calculated as the mean value of qualities on validation sets over all iterations of described procedure.

Another source of bias in the quality estimator is mismatch of target distributions in the training and validation samples. Because the splits in the standard cross-validation procedure are generated randomly the weights of positive samples (in binary classification tasks) in the trained model may differ from the data on which quality control is performed. To avoid this effect stratified sampling scheme (preserving the proportions of the target) is usually applied.

## 3 Related work

### 3.1 Early stopping

Early stopping is a task of controlling the learning process and interrupting it to avoid unnecessary boosting steps, which increase complexity of the model and can lead to overfitting. Since the number of learning steps is directly connected to the complexity of the model, larger ensemble sizes lead to models of smaller bias but larger variance (bias–variance tradeoff). One of the ideas proposed in the literature (Chang et al. (2010), Mayr et al. (2012)) is to penalize the complexity of the models, e.g., via AIC-based methods by approximating the ensemble's degrees of freedom. Some works use generalization bounds of the algorithm employing VC-dimension (Freund & Schapire (1997)), Rademacher complexity (Cortes et al. (2019)), or in the PAC setting (Yao et al. (2007), Wei et al. (2017)). These methods do not require separate validation control, but in most cases, are not applicable in real-world tasks since the obtained bounds are distribution-agnostic. Therefore the approximations are very rough.

Standard approaches of early stopping mentioned in most of the well-known implementations of GB utilize the simple "waiting" idea: if the validation quality does not change for some "reasonable" number of iterations, then the training must be stopped (see, e.g., Click et al. (2016)). It is important to note that this kind of early stopping may be performed simultaneously with the boosting learning

procedure step-by-step so that the training is stopped at the same time when a specific criterion is met. Unfortunately, this method has nothing to do with the double–descent problem (Belkin et al., 2019), and the choice of the required number of waiting rounds remains at the researcher's discretion based on experience or heuristic assumptions.

## 3.2 ENSEMBLE PRUNING

Pruning often refers to various techniques for compressing models for more efficient storage and inference compexity. Several papers addressed this topic in ensembles, since they usually contain a large number of models. For example, some studies address the problem of adaptive online pruning in Multiple Classifier Systems setting, where classifiers are learned independently like in bagging, see Cruz et al. (2015) and (Cruz et al., 2018) for review; (Oliveira et al., 2017) and (Hernández-Lobato et al., 2008) propose an instance–wise pruning methods that allows to halt some models at inference time, while in (Soto et al., 2014) both static (training time) and dynamic (inference time) pruning in AdaBoost are investigated. In this paper, we consider Gradient Boosting, where a crucial regularisation technique is early stopping based on (cross–)validation.In practice and in other works on pruning (e.g. (Fan et al., 2002)), we can see that it also have regularizing effect, it is often noticeable that we can get the ensemble with significantly better quality. The latter can be provided by eliminating the flaws of the model obtained due to the greedy learning algorithm.

The classic work on this task (Margineantu & Dietterich, 1997) compared five different pruning methods applied to boosting algorithm. In most cases pruned models were able to maintain and increase the original quality with a moderate reduction in the size. Most of modern pruning techniques are based on the fact that similar learners in the ensemble duplicate the information about the dataset, so they can be eliminated from a model sequence (Cavalcanti et al., 2016; Li et al., 2012). There also have been tries to formulate ensemble pruning as an optimization problem and apply genetic algorithms (Zhou & Tang, 2003) or semi-definite programming (Zhang et al., 2006) to find a solution.

In this paper we follow a standard pruning scheme described in (Margineantu & Dietterich, 1997): shrink the model to the first $M$ learners, giving the best validation score. But unlike all previous works on gradient boosting, instead of a universal constant, we strive to select this number adaptively for different regions at training time, taking into account the distribution of the training data.

## 4 ADAPTIVE EARLY STOPPING

In Section 2, we have described the boosting ensemble in the form $F_B(\boldsymbol{x}) = \sum_{i=1}^{B} \alpha f_i(\boldsymbol{x})$, where $B$ is the total number of models in the ensemble. When we apply $k$-fold cross-validation scheme to determine the optimal number $M$ of addends, we get $k$ different $B$-sized models $\{F_B^j\}_{j=1}^k$:

$$F_B^j(\boldsymbol{x}) = \sum_{i=1}^{B} \alpha f_i^j(\boldsymbol{x}),$$

learned by $k$ training sets $\{\mathcal{S}_{-j}\}_{j=1}^k$.

The $j$-th cross–validation step provides quality estimator $\boldsymbol{l}_j = (l_j^{(1)}, l_j^{(2)}, ..., l_j^{(B)})$ obtained by applying all the prefixes of the model $F_B^j$ to validation set $\mathcal{S}_j$. In other words,

$$l_j^{(b)} = \frac{1}{|\mathcal{S}_j|} \sum_{(\boldsymbol{x},y) \in \mathcal{S}_j} L\left(F_b^j(\boldsymbol{x}), y\right),$$

where $F_b^j = \sum_{i=1}^{b} \alpha f_i^j$. The final estimator $\boldsymbol{l} = \frac{1}{k} \sum \boldsymbol{l}_j$ is further used to define estimated value of $M$ as $\hat{M} := \arg\min_{1 \le i \le B} l^{(i)}$. The model shrinked to the first $\hat{M}$ iterations provides an estimator with the test quality close to $\min_M \mathbb{E}_{(\boldsymbol{x},y) \sim P}[L(F_M(\boldsymbol{x}), y)]$. But the problem is that the desired estimator should be selected with the goal to be an approximation to $\mathbb{E}_{(\boldsymbol{x},y) \sim P} \min_M [L(F_M(\boldsymbol{x}), y)]$, due to an obvious inequality:

$$\mathbb{E}_{(\boldsymbol{x},y)\sim P} \min_M [L(F_M(\boldsymbol{x}), y)] \leq \min_M \mathbb{E}_{(\boldsymbol{x},y)\sim P}[L(F_M(\boldsymbol{x}), y)]. \tag{3}$$

This simple mathematical fact convinces us that the existing pruning scheme is being used ineffectively. Adaptive selection of numbers for specific examples can achieve better quality by eliminating the theoretical gap given by inequality 3. In the following sections, we describe possible approaches to adaptive iteration count selection and evaluation of its effect.

### 4.1 MAIN IDEA

Suppose the input space $\mathcal{D}$ is divided into $C$ disjoint regions $(\mathcal{D}_1, \mathcal{D}_2, ..., \mathcal{D}_C)$ in such a way that all samples in $\mathcal{D}_i$ are close to each other in some sense (they follow the same latent distribution or geometry). Note that this partition is unrelated to the split induced by cross-validation, since the latter split is done randomly and there is no reason to expect closeness of samples inside a single fold. We assume that $(\mathcal{D}_1, \mathcal{D}_2, ..., \mathcal{D}_C)$ is a clustering in the sense that data points of the same cluster $\mathcal{D}_i$ behave similarly during the procedure of training an ensemble. In particular, the optimal number of boosting iterations $\hat{M}_i$ estimated for $\mathcal{D}_i$ may differ a lot from the one estimated for $\mathcal{D}_j$. Therefore, by analogy with the inequality 3, we can conclude that ensemble size selection based on partition $\mathcal{D}$, where the size is chosen individually for each cluster $\mathcal{D}_i$, can have better quality compared to one "universal" common size:

$$\mathbb{E}_P \min_M [L(F_M(\boldsymbol{x}), y)] \leq \mathbb{E}_{\mathcal{D}_i \sim \mathcal{D}} \min_M \mathbb{E}[L(F_M(\boldsymbol{x}), y)|\mathcal{D}_i] \leq \min_M \mathbb{E}_P[L(F_M(\boldsymbol{x}), y)]. \tag{4}$$

Setting $C = n$ may achieve the theoretical lower bound on the left-hand side of Equation 4. However, the size $M$ of the ensemble will be optimized based on the empirical estimation of the loss, and the growth in $C$ is accompanied by the growth of the variance of this estimation for each region $\mathcal{D}_i$. So the number of regions should be selected reasonably (we discuss it further in the text).

The upper-level training algorithm consists of 4 steps: 1) Cross-validated training of $k$ models; 2) Distributed-based partition $(\mathcal{D}_1, \mathcal{D}_2, ..., \mathcal{D}_C)$ of the sample space; 3) Selecting optimal number of iterations $(\hat{M}_1, \hat{M}_2, ..., \hat{M}_C)$ for each region obtained on the step 2; 4) Retraining the model on the whole training data. The formal description is presented in the Algorithm 1.

The framework described above has two additional hyperparameters: number of clusters $C$ and the minimal size of a cluster (optional), both can be tuned. In Section 4.4, we describe a tuning approach, which makes a minor contribution to the total computational cost comparing to the ensemble training as our tuning method does not require any retraining.

---

**Algorithm 1** Adaptive stopping procedure

    **Input:** $\mathcal{S} = (\boldsymbol{X}, \boldsymbol{y})$
   $folds \leftarrow (\mathcal{S}_1, \mathcal{S}_2, ..., \mathcal{S}_k) \leftarrow CvSplit(k, \mathcal{S})$
   $cvPredictions \leftarrow CvPredict(folds)$
   $partition \leftarrow (\mathcal{D}_1, \mathcal{D}_2, ..., \mathcal{D}_C) \leftarrow GetPartition(\mathcal{S})$
   $bestIterations \leftarrow EstimateBestIterations(folds, cvPredictions, partition)$
   $finalModel \leftarrow Train(\boldsymbol{X}, \boldsymbol{y}, partition, bestIterations)$
   **return** $finalModel$

---

### 4.2 UNSUPERVISED PARTITION

As it was mentioned in Section 4.1, the partition should reflect the internal structure of the data to be sophisticated enough to select a proper number of models. Let us use a reasonable assumption that observations that are close in the feature space are also close in their properties. Then we can use one of clusterization algorithms (e.g., KMeans (Lloyd, 1982), EM (Dempster et al., 1977), agglomerative method (Sibson, 1973)) to get data partition (function $GetPartition$ in Algorithm 1).

It is essential to preserve the initial geometry of the input space since most of the modern implementations of Gradient Boosting use Decision Tree (Breiman et al., 2017) as a base learner. Decision

Tree constructs piecewise-constant approximations at each step, and it is more likely for close instances to get into the same leaves during training and inference, so they tend to fit equally. The unsupervised partition method allows controlling the number of partition regions and their sizes via setting the desired number of clusters and minimal samples count in each cluster.

This method being applied to real data exhibits several disadvantages. First, clustering does not work well with data in which non-numeric categorical features are present. Numeric encoding of high-cardinality categorical features leads to sparse input space and dramatically affects clusterization's capacity. Second, unsupervised partition does not consider the labels of the data points, although they may contain valuable information about the required number of boosting steps. Last, some advanced clusterization algorithms require high computational costs what can become a bottleneck when training a model.

### 4.3 TREE-BASED PARTITION

To avoid issues described in the previous paragraph, the partition unit (function $GetPartition$ in Algorithm 1) should be scalable, interpretable in terms of built subspaces, and tolerant to heterogeneous feature input. We find the Decision Tree model to be a suitable candidate since it satisfies all the listed properties: training algorithm is parallelizable and not memory consuming (Sharp, 2008), cluster manifolds are similar to the ones built by base learners, there are efficient categorical feature supporting methods (Prokhorenkova et al., 2017).

The partition procedure boils down to training a single decision tree on the initial training samples and targets. Then, we denote each leaf as a separate cluster of the data forming the partition. Since the tree learning process utilizes both geometry of feature space and target distribution in leaves to split the data, this method is encouraged to find the regions similar by feature representation and label. In other words, it uses all available information about the data.

This partition tree may be trained separately from the primary boosting model as well as be the first booster in the ensemble. The latter means that this step does not affect the training time at all. However, since Gradient Boosting usually consists of hundreds and thousands of trees, the effect on time costs of using a separate partition model is negligible.

The number of clusters and cluster sizes can be controlled via setting an appropriate number of leaves in the tree and minimal leaf size.

### 4.4 VALIDATION PROTOCOL

It is still an open question how to select the values $\hat{M}_1, ..., \hat{M}_C$ for each cluster (function $EstimateBestIterations$ in Algorithm 1). Also, adopting new options to any machine learning algorithm raises questions on the limits of applicability and the possibility of extending experimental results and theoretical calculations to real problems and data. New hyperparameters, as a rule, make training procedure more complex and increase the tuning time due to enlarged hyperparameter search space. In this section, we demonstrate that evaluation and tuning of the proposed method require only one model training step and one inference. All the rest of the work can be done just with the help of precalculated cross-validated predictions, so it is cheap to determine the optimal parameters and estimate the possible effect on the quality of the final model.

Let us denote $\mathcal{D}_{i,j} = \mathcal{D}_i \cap \mathcal{S}_j$ the set of observations from the $j$-th fold belonging to the cluster $\mathcal{D}_i$ and $n_{i,j} = |\mathcal{D}_{i,j}|$. Naive approach (Algorithm 2) of evaluation consists of applying cross–validation model trained on the sample $\mathcal{S}_{-j}$ to the validation set $\mathcal{S}_j$ for any $j$, obtaining quality estimators $l_{i,j}$:

$$l_{i,j}^{(b)} = \frac{1}{n_{i,j}} \sum_{(\boldsymbol{x},y) \in \mathcal{D}_{i,j}} L\left(F_b^j(\boldsymbol{x}), y\right).$$

The resulting estimator $\boldsymbol{L}_i$ for each cluster $i$ is a weighted sum of corresponding cluster estimators over all folds:

$$L_i^{(b)} = \frac{\sum_{j=1}^{k} n_{i,j} \cdot l_{i,j}^{(b)}}{\sum_{j=1}^{k} n_{i,j}},$$

then $\hat{M}_i := \arg\min \boldsymbol{L}_i$ and the cross–validation score of cluster $i$ equals to $\min \boldsymbol{L}_i$. The total complexity of the described procedure is $O(C(B + k) + nB)$, which is meager compared to the ensemble training complexity, which is at least $O(nmdB)$ (Friedman, 2001) (for $m$ binary features and trees of depth $d$).

---

**Algorithm 2** Best Iteration Selection

> **procedure** ESTIMATEBESTITERATIONS($folds, cvPredictions, partition$)
>  **for** $\mathcal{D}_i \leftarrow partition$ **do**
>    $\boldsymbol{L}_i \leftarrow \vec{0}$                                 ▷ vector of $B$ zeros
>    $n_i \leftarrow 0$
>    **for** $\mathcal{S}_j \leftarrow folds$ **do**
>      $\mathcal{D}_{i,j} \leftarrow \mathcal{D}_i \cap \mathcal{S}_j$
>      $n_{i,j} \leftarrow |\mathcal{D}_{i,j}|$
>      $\boldsymbol{L}_i \leftarrow \boldsymbol{L}_i + Eval(cvPredictions[\mathcal{D}_{i,j}]) \cdot n_{i,j}$     ▷ elementwise vector sum
>      $n_i \leftarrow n_i + n_{i,j}$
>    **end for**
>    $\boldsymbol{L}_i \leftarrow \boldsymbol{L}_i / n_i$
>    $M_i \leftarrow \arg\min \boldsymbol{L}_i$
>  **end for**
>  **return** $\{M_i\}$
> **end procedure**

---

Obviously, the quality assessment obtained in the way described above is biased and always gives an optimistic estimate. In particular, it is impossible to use this quality estimator to determine an optimal number of clusters, as it always monotonically increases with finer clustering. For a more accurate assessment of generalization ability, we suggest using the following cross-validation evaluation procedure in Algorithm 3, which does not allow target leakage and strong bias. For each fold $\mathcal{S}_q$, we compute an optimal stopping moment for cluster $i$ by averaging evaluation metrics for all observations from cluster $i$ that do not belong to fold $\mathcal{S}_q$. More formally, we compute $\boldsymbol{L}_{i,-q}$ as

$$L_{i,-q}^{(b)} = \frac{\sum_{j \neq q} n_{i,j} \cdot l_{i,j}^{(b)}}{\sum_{j \neq q} n_{i,j}},$$

by applying $EstimateBestIterations$ (Algorithm 2) to all folds except the $q$-th one (ignoring $\mathcal{S}_q$ from $folds$). After this step, we have $(\hat{M}_1^q, ..., \hat{M}_C^q)$ estimated on $\mathcal{S}_{-q}$. Then we use $\mathcal{S}_q$ as a set validating the quality of predicted $(\hat{M}_1^q, ..., \hat{M}_C^q)$. After averaging the obtained results over folds $\mathcal{S}_q$, we get a more accurate estimation of the quality of clustering, which is used to select the number and size of clusters and to estimate the possible profit of applying adaptive stopping procedure, all this with a minor additional time consumption relative to the training time of the ensemble model. There is still some bias because fold $S_q$ is used both to train models applied to $\mathcal{S}_{-q}$ and to estimate the performance of stopping points. However, the desired property of not using the same set for both tuning and evaluating $M$ is satisfied and allows us to get useful estimations.

## 5 EXPERIMENTS

In this section, we perform numeric experiments, analyze the effectiveness of the proposed framework, and validate statements made in Section 4. We take a popular open–source Gradient Boosting library, CatBoost (CatBoost, 2017). It is known for achieving SOTA results on a large number of

---

**Algorithm 3** Evaluation Procedure

**procedure** EVALUATE($folds, cvPredictions, partition$)
    **for** $\mathcal{S}_q \leftarrow folds$ **do**
        $\{M_i^q\} \leftarrow EstimateBestIteration(folds \setminus \mathcal{S}_q, cvPredictions, partition)$
        $predictions_q \leftarrow cvPredictions[\mathcal{S}_q]$
        **for** $\mathcal{D}_i \leftarrow partition$ **do**
            Shrink($predictions_q[\mathcal{S}_q \cap \mathcal{D}_i], M_i^q$)
        **end for**
        $L_q = Eval(predictions_q)$
    **end for**
    **return** $Mean(\{L_q\})$
**end procedure**

---

Table 1: Datasets

|  | Adult | Amazon | KDD Upselling | Kick | KDD Internet | Click | Higgs | Marketing | Default | HEPMASS |
|---|---|---|---|---|---|---|---|---|---|---|
| **#samples** | 49K | 33K | 50K | 73K | 10K | 400K | 11KK | 45K | 30K | 840K |
| **#features** | 15 | 10 | 231 | 36 | 69 | 12 | 28 | 16 | 23 | 25 |

benchmarks (Bentéjac et al., 2021) with the use of default settings and an efficient integrated categorical feature handler. We train each model for $B = 5000$ iterations with the learning rate set in such a way that the cross validated optimal point is close to the 2500-th iteration to ensure convergence of the training proceess. Datasets used in this investigation and their properties are listed in Table 1, their links can be found in references. Most of them are taken from the list of benchmarks of the original CatBoost paper (Prokhorenkova et al., 2017) and the proposed tuned hyperparameters from the paper were used.

We hold out 20% from each dataset for the test. The 5–fold stratified cross-validation is utilized to determine the optimal stopping moment. We use the standard pruning algorithm as a baseline and compare it with the method proposed in Section 4. The clustering is performed by training a separate non-symmetric decision tree ("Lossguide" training policy) on the train data and initial labels, where each leaf is an individual cluster. To control cluster count and minimal cluster size, we utilize "num_leaves" and "min_data_in_leaf" decision tree parameters respectively.

**Does it matter to select a different number of iterations for different regions?** To address this question, we train boosting and clustering models on the train data, apply the model to the test set and evaluate metrics after each prediction step (iteration). Then we compare the universal optimal step number, calculated as the minimum point of the whole test data loss, and adaptive by independently calculating the optimal size for each cluster. The described procedure was carried out 20 times for different train/test splits. If the assumption from Section 4 is false, we would see that step numbers calculated for clusters are not diversified a lot and distributed close to the general optimal iteration. Nevertheless, in reality, we face the situation when the best iteration differs from the ones obtained for each cluster. For example, Figure 3 in Appendix (each line is a different train/test split) demonstrates evaluation history for the whole dataset and its two clusters with discrepant optimal stops.

These observations motivate our research and confirm the inefficiency of the classical approach. Also, it is interesting to note that many clusters and instances are well-trained long before the optimal moment is reached. Therefore, subsequent iterations work in vain, wasting time on their training, not to mention that these examples add additional noise to the predictions for the remaining examples. It is also easy to notice that a significant part of the clusters has the best iteration value close to the size of the ensemble $B$. The latter means that there are plenty of underfitted data points ($B$ steps is not enough) that can not be caught by the classical method. However, the partitioning proposed in this article allows them to be detected and trained for an additional number of steps.

**Does the validation protocol proposed in Section 4.4 have good generalization ability?** For this investigation we applied naive validation control, described in Section 4.4, and advanced evaluation procedure (briefly in Algorithm 3) to every dataset. As we can see from Figure 1 and Figure 2 naive

Table 2: Quality estimation, 0-1 loss / logloss, relative error change

|  | Adult | Amazon | KDD Upselling | Kick | KDD Internet |
|---|---|---|---|---|---|
| **Baseline** | 0.1264 / 0.2723 | 0.0447 / 0.1400 | 0.0494 / 0.1666 | **0.0496** / 0.2857 | 0.1004 / 0.2202 |
| **Adaptive pruning** | **-0.24% / -0.24%** | **-1.37% / -0.53%** | **-0.20% / -0.10%** | +0.11% / **-0.19%** | **-2.46% / -0.52%** |
|  | Click | Higgs | Marketing | Default | HEPMASS |
| **Baseline** | **0.1564** / 0.3916 | 0.2364 / 0.4810 | 0.0926 / 0.1937 | 0.1865 / 0.4327 | 0.1258 / 0.2768 |
| **Adaptive pruning** | +0.04% / **-0.03%** | **-0.14% / -0.14%** | **-2.27% / -0.71%** | **-2.50% / -0.07%** | **-0.17% / -0.16%** |

validation protocol monotonically decreases with the number of clusters, as it was expected, and it gives no insight about the optimal cluster count and possible improvement compared to the baseline. In contrast, the quality estimation produced by the advanced approach is highly correlated with test quality. The quality patterns for test and validation are repeated, and there is an opportunity to make an informed choice of a number of clusters and other parameters affecting clustering.

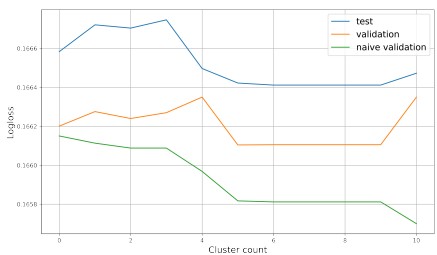

Figure 1: Upsel, validation

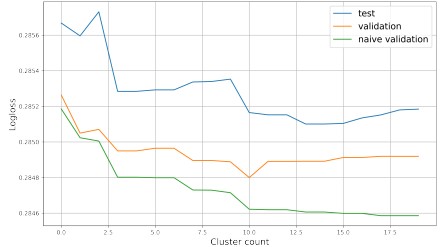

Figure 2: Kick, validation

**Does the proposed algorithm help to increase the quality of boosting models?** In this paragraph, we carry out an extensive search of the best partition in terms of two loss metrics (lower is better): Logloss and 0-1 loss. The number of clusters is tuned according to the procedure from Section 4.4. Then we find the optimal iteration count for each cluster and apply the corresponding number of trees (boosters) to each test sample (as in Algortithm 1). The comparison with the baseline is presented in the Table 2. The results show the superiority of the proposed technique over the classic early stopping on most settings. The improvements are significant according to Wilcoxon signed-rank test with $p - value \ll 0.001$, except for datasets Click and Kick. From this, we can conclude that modern Gradient Boosting implementations do not use the full power of the models, limiting themselves to the shared stopping moment for all examples. At the same time, the personalized selection of this parameter allows significant improvements in the algorithm's performance. In this paper, we select the optimal number of clusters under the assumption of using a separate clustering tree. However, at the same time, we firmly believe that the optimal construction of the clusters themselves (for example, taking into account the learning history of the instance) can bring even greater success.

## 6 CONCLUSION AND FUTURE WORK

In this paper, we discovered a problem of ensemble pruning previously uncovered in the literature. We discussed possible problems that the simultaneous stopping rule brings to the modern boosting models and proposed a cluster-based framework of early stopping that can be directly applied to any implementation of Gradient Boosting (and possibly other ensemble methods) without harming its quality and training/inference time. We proposed an evaluation protocol for our method, so it is simple and at the same time computationally cheap to determine whether the adaptive stopping works well for any particular data. Our experiments with the well-known implementation of boosting demonstrate the validity of the assumptions and conclusions made in the paper and great potential for applications and further research since this work still uncovers many problems.

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

## 7 APPENDIX

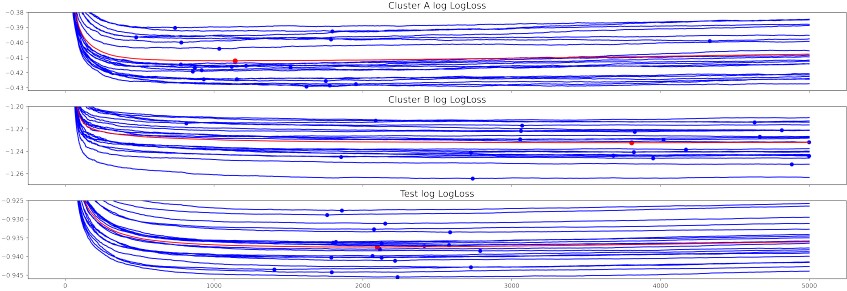

Figure 3: Click, evaluation history. Blue lines indicate runs for different train/test splits, red lines are averages over all runs. Dots specify the minimum.

