# OpenReview forum: "Learn Together, Stop Apart: a Novel Approach to Ensemble Pruning"
_ICLR.cc/2022/Conference — ICLR 2022 Submitted_

### Official Review · Reviewer_xGqg · 2021-11-02

**Correctness:** 4
**Technical Novelty And Significance:** 3
**Empirical Novelty And Significance:** 3
**Recommendation:** 5
**Confidence:** 5

**Main Review:**

**Strong points:**
- Very simple method, with easy implementation (making its reproduction straightforward);
- Elegant idea for incorporating the approach within a cross-validation procedure.
- Strong results (albeit in a short amount of datasets).

**Weak points:**
- Very limited experimental analysis. For addressing that I would recommend:
    - You should try it on dozens of datasets, given that the method is allegedly very fast.
    - You should compare your approach with several methods for hyperparameter optimization, not only with the naive CV procedure. I am curious to see whether a more sophisticated hyperparameter tuning approach can outperform you simple approach, and if so, with which efficiency it would do it.
    - You should employ statistical tests for better assessing the statistical significance of results. In particular, there are several nonparametric tests that can help you pointing out significant differences, and also post-hoc methods for pointing out pairwise differences for the case of multiple methods being executed over multiple datasets.

**Questions:**
- You say you define the number of clusters (leaves in the decision tree) according to the procedure described in Section 4.4. It is not clear to me which procedure is that. Do you mean running Alg. (3) several times with distinct number of clusters, and using the one with the best results? How exactly did you do that? Since defining the number of leaves in decision-tree induction is not straightforward (you need to try and control that via tree height or minimum number of instances per leaf), I would guess that this step is a little bit more complicated than you make it appear (and also, the increased computational cost of running your entire procedure - which has a non-negligible cost itself - multiple times). I think that is a vital part of the method whose discussion is not done at all in the paper.
- The "shrink" method in Alg. (3) is not detailed at all, so I am assuming it means getting the predictions of the full model executed over fold $S_q$ and "cutting off" the results that use more than $M_i^q$ models, would that be correct? I am sorry for the confusion, but that is not totally clear to me.
- In inference (test) time, I would assume you need to see in which cluster the new instance falls on, and then using the number of models defined for that cluster, is that also correct? I ask because there is no mention on that at all in the paper.

**Summary Of The Paper:**

The paper proposes a novel method to set the optimal ensemble size in gradient boosting. In particular, the authors propose an adaptive strategy that sets distinct ensemble sizes for different regions of the input space. For that, they propose dividing the input space in coherent regions (whose instances are similar both in terms of features and labels) and estimating the optimal ensemble size within those regions. For clustering data, they propose using a decision tree induced from the entire training set, and the leaves of the trees are the clusters that comprise the data partition.
They show the results of both a biased and a less-biased estimator for finding out the optimal ensemble size per region, and they compare their findings with the traditional strategy of simply pruning the ensembles based on a single optimal number of learners estimated from a cross-validation procedure. Results in 6 public datasets show that in at least 4 of those datasets the method seems to provide better results.

**Summary Of The Review:**

Paper with a nice, elegant, and very simple idea for adaptively generating different number of ensemble sizes throughout the input space, in an attempt to generating better results.
The weak point is that the experimental analysis is not up to the standards that one would expect here, as I would have expected the method to be tested over dozens of datasets, with a proper analysis against other hyperparameter optimization approaches, and statistical tests to validate the significance of the results.

---

> ### Author Response · Authors · 2021-11-23
> **Thank you for your comments!**
>
> Thank you for your comments and for highlighting the weak points of our paper. We will do our best to strengthen it.
>
> In particular, we provide the experimental part with additional datasets and statistical tests proving the significance of obtained improvements, see revised version.
>
> - You should compare your approach with several methods for hyperparameter optimization, not only with the naive CV procedure. I am curious to see whether a more sophisticated hyperparameter tuning approach can outperform you simple approach, and if so, with which efficiency it would do it.
>
> Please, note that hyperparameter optimization is an orthogonal way to improve model quality. CV procedure is a standard framework, which is needed to estimate the value of the optimised black box (test model quality). It is usually combined with different methods of hyperparameter optimization: grid, random search, TPE, GP, and so on. In our experiments, we used SOTA hyperparameters fitted individually in previous works to each individual dataset. Unlike hyperparameter optimization, which needs multiple runs of CV procedure, our method makes negligible additional computational overhead.
>
> Answers to your questions:
> 1) Yes, we run Algorithm 3 for each clustering to select the best one. We control the number of clusters via min_data_in_leaf and num_leaves or depth (for symmetric trees) parameters. Deep trees can be reused to obtain less fine partitioning with the help of non-terminal nodes. We agree that the tuning procedure should be discussed more in the paper, and we will consider doing it in the next revision.
> 2) Yes, you are right. 'Shrink' means discarding predictions of all trees except the first M.
> 3) This is our flaw. Yes, at inference time, the test set is divided into clusters according to the clustering tree, and then we apply the desired number of trees to each cluster.

---

> > ### Comment · Reviewer_xGqg · 2021-11-29
> > **Comments after reading the novel version**
> >
> > Dear authors,
> > Thank you for your addressing the issues I've mentioned, and for the revised version of the manuscript.
> > Certainly 10 datasets is better than 5, though I was hoping for a bit more. =)
> > Nevertheless, nice effort, conclusions seem to hold for the novel datasets. I would suggest inserting the actual values of the evaluation measures instead of relative gains, though, for completeness.
> >
> > Regarding the fact that hyperparameter optimization is orthogonal to you work, I would partially agree with you on that. But first, I would ask you to clarify the matter of the baseline hyperparameters used for each dataset you tested on. You affirm they are the "SOTA hyperparameters*, but what does that really mean? Were they optimized by someone other than you? How was such an optimization performed? Was the method used different per dataset?
> > I think it is *really important* that you clarify this matter before the decision deadline.

---

> > > ### Author Response · Authors · 2021-11-30
> > > **Answer to comments**
> > >
> > > By “the SOTA parameters” we mean the hyperparameters selected in previous works using CatBoost as a boosting implementation. Among them are the papers from NeurIPS 2019 (MVS in GBDT, Section 5. the beginning of p.8) (https://papers.nips.cc/paper/2019/hash/5c8cb735a1ce65dac514233cbd5576d6-Abstract.html) and NeurIPS 2018 (original CatBoost paper, Section 6 and Supplemental, Section D)(https://papers.nips.cc/paper/2018/hash/14491b756b3a51daac41c24863285549-Abstract.html). The latter paper contains a description of the selected parameters for datasets and a protocol for their configuration. In particular, 5-Fold cross-validation and 50 iterations of hyperopt were used to tune the parameters. We left these settings untouched, except for the learning rate, which we set with the purpose of making the global optimal point close to the 2,500th iteration (so that the final model allows both under- and overfitted samples).
> > > The more detailed information and the code may be found at https://github.com/catboost/catboost/tree/master/catboost/benchmarks/quality_benchmarks

---

### Official Review · Reviewer_meXP · 2021-11-02

**Correctness:** 3
**Technical Novelty And Significance:** 3
**Empirical Novelty And Significance:** 3
**Recommendation:** 6
**Confidence:** 4

**Main Review:**

The idea of partitioning the input space and using a varying number of models in the ensemble makes a lot of sense to me and is novel, to the best of my knowledge. Gradient boosting and similar techniques do indeed remain widely used and very competitive for small-to-medium dimension tabular data, so the practical significance of the method is clear. Although some minor edits are needed, the presentation of the technique is clear for the most part.

My main concern is that the number of benchmark datasets is smaller than I would have liked. Given the option nowadays to rent computing power temporarily via the cloud, it's not unreasonable to expect much more than 6 benchmarks to be tested for a method like this. For instance, I once refereed a paper for another conference which tested their method on 40 benchmarks- and (although I liked the paper), the paper got rejected ! I am not saying 40 is required, but maybe 12 or 15 benchmarks would make me more comfortable with the results than just 6.

With that said, the method is elegant and intuitive and widely applicable, so I don't mind at all if this paper is accepted. The small number of benchmarks is the reason I can only give it a 6.

One major suggested edit: the author use the term 'object' in a non-standard way, to refer to what are normally referred to as examples, ( or data points, instances, observations, etc), i.e. the labelled (x,y) pairs used to train a supervised learning model. I suggest choosing a term other than 'objects'.

One minor complaint: on page 6, it is claimed that cluster surfaces are hard to validate. In fact, clustering can be evaluated out of sample much in the same way as supervised learning, by looking e.g. at likelihood on a holdout sample, so as to choose an optimal number of mixtures in a mixture of gaussians. It is also possible to construct objective functions for something like k-means which trade off K with distance of each data point to its cluster mean. Nonetheless, it does make sense to me to partition with a decision tree, so I am not bothered by this misstatement much.

A few other suggested edits:

Page 3 The early stopping -> early stopping

Page 8 Does the validation protocol proposed in Section 4.4 has good generalization ability? Has -> have

**Summary Of The Paper:**

An enhancement of the popular gradient boosting method is presented wherein the input space is subdivided into regions and the cross-validated optimal number of trees to include in the ensemble is chosen on a per-region basis rather than using a single global number of trees for the entire input space. Consistently superior performance relative to standard gradient boosting across 6 benchmark datasets is reported.

**Summary Of The Review:**

A method for partitioning the input space for a gradient boosting algorithm and choosing a different number of trees to be included in the ensemble in a partition-dependent way is presented. Improvements on 6 benchmark datasets relative to standard gradient boosting are reported. The method is intuitive and elegant but the number of benchmark datasets is not as large as it could be, and for that reason, the paper gets a 6 (marginal accept).

---

> ### Author Response · Authors · 2021-11-23
> **Glad you liked the method!**
>
> Thank you for the review and your valuable comments, we fixed the paper according to them. We are glad that you liked the work and the proposed method.
>
> We do not indicate this in our article, which is a significant omission, but the choice of the benchmarks is not accidental. We use the data directly from the papers "Catboost: unbiased boosting with categorical features" and "Minimal variance sampling in stochastic gradient boosting" to perform a fair comparison and demonstrate the quality improvement on a specific algorithm tuned to achieve the best results on these benchmarks. Nevertheless, we did not mind considering some more data, and we added some new results in the revised version.
>
> - ​​One minor complaint: on page 6, it is claimed that cluster surfaces are hard to validate.
>
> We corrected that statement, thank you for the point!
>
> The remark about the term "object" is reasonable! We will fix it.

---

> > ### Comment · Reviewer_meXP · 2021-11-29
> > **Thanks for making the edits and adding new results**
> >
> > I will keep my score the same, but I do think the paper has benefited from the changes.

---

### Official Review · Reviewer_EFwa · 2021-11-04

**Correctness:** 3
**Technical Novelty And Significance:** 2
**Empirical Novelty And Significance:** Not applicable
**Recommendation:** 5
**Confidence:** 4

**Main Review:**

Adapting the ensemble size in an instance-wise fashion is an interesting problem. It is not new however, unlike what is claimed in the paper. This is addressed at least in these papers:

[1] Hernández-Lobato D, Martínez-Muñoz G, Suárez A. Statistical instance-based pruning in ensembles of independent classifiers. IEEE Trans Pattern Anal Mach Intell. 2009 Feb;31(2):364-9.
[2]  Dayvid V.R. Oliveira, George D.C. Cavalcanti, Robert Sabourin, Online pruning of base classifiers for Dynamic Ensemble Selection, Pattern Recognition, Volume 72, 2017.
[3] V. Soto, S. García-Moratilla, G. Martínez-Muñoz, D. Hernández-Lobato and A. Suárez, "A Double Pruning Scheme for Boosting Ensembles," in IEEE Transactions on Cybernetics, vol. 44, no. 12, pp. 2682-2695, Dec. 2014, doi: 10.1109/TCYB.2014.2313638.
[4] Rafael M.O. Cruz, Robert Sabourin, George D.C. Cavalcanti, Dynamic classifier selection: Recent advances and perspectives, Information Fusion, Volume 41, 2018, Pages 195-216.

The method proposed in the present paper, based on clustering and cross-validation, seems however novel and significantly different from these works, which perform pruning in an online way, i.e. when a prediction needs to be done. In addition, the motivation of most of these works is also to reduce memory and computing times more than improving accuracy. I think the related work discussion should nevertheless includes these approaches.

The proposed method is very straightforward. It makes sense overall but I don't totally buy some of the arguments that are given in Section 4 to motivate it:
- First, the discussion focuses only on the impact of pruning on predictive performance. While pruning can be very useful to reduce storage requirement or computing times at inference (and this is angle adapted in, e.g., [3] above), I'm not sure that it is that useful for improving predictive performance. Boosting has been shown to be quite robust with respect to the ensemble size (if the learning rate is small enough) and actually, experiments in Figure 1 and 2 confirm this fact, since most error curves are monotonically decreasing (I see only one curve that really significantly increases at some point). So, I was not expecting a priori a huge improvement in terms of predictive performance.
- Second, some statements to motivate the idea of clustering are not really supported either by a theoretical or an empirical analysis (although I agree that they make sense intuitively). If you want to show that "It is essential to preserve the initial geometry" and to exploit the labels at the clustering stage, then you should provide experiments to show that not doing so is indeed detrimental (by using e.g. unsupervised clustering).
- Third, the fact that the clustering is carried out without consideration of the boosting model makes the approach also suboptimal by design. There is no guarantee that the clustering will be optimal when it comes to tune the ensemble size in each cluster.

The idea of the cross-validation approach based on a single model (per fold) is sound and very relevant to reduce the computational cost of the approach. Note however that the cost is still important with respect to no pruning at all since it requires to grow k+1 boosting ensembles instead of 1. But I agree that there is a negligible overhead with respect to tuning globally the ensemble size.

The empirical validation is carried out correctly from a statistical point of view. I find however that the authors overemphasize the significance of the improvement they obtain with their approach. Looking at Table 2, their local pruning technique brings an improvement of less than 1% in terms of 0-1 loss on four problems out of six and on the other two problems, the difference remains very small. I'm not sure that such level of improvement is actually worth the effort if one is only interested in predictive performance. These results are also obtained with a single dataset split (if I understand correctly). Given how small the difference is, I think it would have been important to repeat the experiment several times with different splits to get standard deviations and maybe also to carry out a statistical test to check whether the improvements are statistically significant.

I don't think either that the experiments provide a satisfactory answer to the first two questions asked in Section 5. As already discussed above, to me, Figures 1 and 2 show that one can not expect strong benefit from the clustering since most error curves are monotonically decreasing. Part of the important diversity in the optimal size in the clusters seems to be due to the fact that long flat regions are observed which lead to an instable position of the optimum. I would have like also a more systematic and quantitative experiment on all datasets to answer the second question about the relevance of the validation protocol. Why not compare the optimum found by this protocol with the "theoretical" optimum found on the test set?

Finally, I'm surprised that the authors don't talk at all about the benefit of pruning on computing times at inference. To me, this is one of the main motivation for using pruning in the context of ensembles. I would be interested to know how global and cluster-based pruning compare from this point of view.

Minor comments:
- In 3.1, the following sentence is unclear: "This method has nothing to do with the double-descent problem". What do you mean?
- It's not clear how you fix the number of leaves in a decision tree? In the standard algorithm, the order in which the nodes are expanded is arbitrary. Do you apply some best-first strategy? Which one?
- If I understand correctly the baseline in Table 2 is the standard globally pruned model. If so, I would like to see also the performance of a unpruned ensemble.
- Only one setting of the boosting algorithm is explored (B=5000, a learning rate of 0.02 and default parameters for the CatBoost model). Yet, one expects that these parameters will have an impact on the performance (it's obvious for B at least). I think that more combinations should be explored to make the conclusions more general.
- Looking at Figure 2, there are more error curves above the average curve than below, which suggest that clusters are potentially unbalanced. It would be interesting to report the size of these clusters.

**Summary Of The Paper:**

This paper proposes to tune the number of models in a boosting ensemble in an instance-wise fashion. The idea is to first cluster the samples using a decision tree and then to tune the size of the ensemble independently for each cluster, instead of doing it globally for all instances. An efficient two-level cross-validation procedure is designed to tune both the number of terms in each cluster and the number of clusters. Experiments are conducted on 6 large-scale problems that show that local pruning brings some improvement with respect to the more standard global pruning technique.


**Summary Of The Review:**

The paper is well written and the proposed method makes sense and is original. But it lacks a bit of theoretical motivation and I'm not convinced of its practical relevance, given the very marginal improvements observed in the experiments.

---

> ### Author Response · Authors · 2021-11-23
> **Thank you for the extensive work!**
>
> Thank you for your valuable comments and suggestions!
>
> - ​​Adapting the ensemble size in an instance-wise fashion is an interesting problem. It is not new however, unlike what is claimed in the paper. This is addressed at least in these papers:...
>
> Thank you for this remark, we corrected our statement in the paper: our work is the first paper that successfully applies instance-wise early stopping to gradient boosting, not to any ensemble method. Thank you also for references to relevant papers, we included them in the updated version of the paper.
>
> - Boosting has been shown to be quite robust with respect to the ensemble size (if the learning rate is small enough) and actually, experiments in Figure 1 and 2 confirm this fact, since most error curves are monotonically decreasing (I see only one curve that really significantly increases at some point).
>
> However Figure 1 and 2 do not show this, there is usually large increase, when you further increase the size of the model. We will provide additional figures to demonstrate this fact. GB is usually applied to noisy data, what lead to significant overfitting effect. Note that early stopping in gradient boosting is a vital regularisation technique used in all practical applications where reaching best possible quality is of large importance.
>
> - I'm surprised that the authors don't talk at all about the benefit of pruning on computing times at inference
>
> In our work, we mainly focus on the regularizing effect of early stopping in Gradient Boosting, as it is one of the main parameters influencing the generalizing ability of the model. We do not position our method to improve training/inference performance in terms of the time complexity since it was invented with other motivation. The approach has a relatively small accelerating and compression potential: in some cases, it allows to detect underfitted regions for which even more decision trees are required in comparison to universal stopping moment. We are sorry for using the term "pruning", if it misleads readers by referring to storage requirements and inference complexity.
>
> - If you want to show that "It is essential to preserve the initial geometry"...., then you should provide experiments to show that not doing so is indeed detrimental
>
> We agree that the paper paid little attention to the clustering scheme. Our experiments showed the superiority of decision tree-based clustering over unsupervised approaches. We will add some discussions and comparisons to the paper.
>
> - "I'm not sure that such level of improvement is actually worth the effort"
>
> We have several points here:
> (1) GB is widely used in large companies including online services. 1% improvement can sometimes mean 1% growth of income. Given the wide applicability of BG, this looks important. Last years, there are not many papers that propose cheap, universal and stable default methods that improve GB.
> (2) We applied our method to the state of the art implementation (Catboost) and took hyperparameters that were already fitted to each individual dataset in previous works.
> (3) We obtained improvements for standard datasets used in previous papers on GB.
>
> - Given how small the difference is, I think it would have been important to repeat the experiment several times with different splits to get standard deviations and maybe also to carry out a statistical test to check whether the improvements are statistically significant.
>
> in the revised version of the paper, we will run the experiments with different splits to show the reproducibility of the results and statistical significance.
>
> Answers to minor comments:
>
> - "This method has nothing to do with the double-descent problem". What do you mean?
>
> We meant "This method cannot tackle the double-descent problem". The problem is that after several iterations of quality decrease of the validation set, the model can converge again to more optimal points. Stopping too early can miss this point.
>
> - It's not clear how you fix the number of leaves in a decision tree?
>
> Most of the modern implementations of decision trees (in particular, included in gradient boosting libraries) allow configuring the maximum number of leaves or the number of layers (for symmetric trees). Primarily, we utilize the 'Lossguide' policy of CatBoost and set an appropriate 'num_leaves' value. It chooses the leaf with the best split score and applies the split to the samples in the current leaf.
>
> - Only one setting of the boosting algorithm is explored (B=5000, a learning rate of 0.02 and default parameters for the CatBoost model).
>
> In fact, we set learning rate in such a way that the cross validated optimal point is close to the 2500-th iteration to ensure convergence of the training proceess. B=5000 was previously shown enough for achieving SOTA result. We agree that other settings are also interesting (for example, with a bounded size of the model) Thank you for all suggestions, we consider providing additional empirical evidences!

---

> > ### Comment · Reviewer_EFwa · 2021-11-30
> > **Comments after revision**
> >
> > I thank the authors for their answer and for addressing some of my concerns (the lack of related references notably). There are still however several issues that have not been addressed satisfactorily and therefore I will keep my score unchanged. I give some further comments below.
> >
> > > However Figure 1 and 2 do not show this, there is usually large increase, when you further increase the size of the model. We will provide additional figures to demonstrate this fact.
> >
> > The new figure 3 is worse than the previous figures 1 and 2 unfortunately. First the figure is barely readable and It's impossible from the figure to determine for which curves there is a significant increase of the error after the minimum (which was my initial point). Again, the position of the minimum might vary only because the curves are very flat. I also don't understand why several train/test splits are now considered. What matters is what happens in the different clusters found for a given train/test split, not how the minimum in a given cluster varies for different train/test splits. I also don't understand what are clusters A and B. The clustering tree is not expected to remain the same for different train/test splits. So, how can you match clusters from different splits? Finally, if this curve is supposed to help answering the question "Does it matter to select a different number of iterations for different regions?", then, it is weird to have used one of the two datasets for which no significant improvement is obtained in Table 2. For this dataset, it is clear that the answer is no.
> >
> > > We agree that the paper paid little attention to the clustering scheme. Our experiments showed the superiority of decision tree-based clustering > over unsupervised approaches. We will add some discussions and comparisons to the paper.
> >
> > I haven't seen any update or experiments related to that in the new version of the paper. This remains an important limitation in my opinion.
> >
> > >	• Given how small the difference is, I think it would have been important to repeat the experiment several times with different splits to get >standard deviations and maybe also to carry out a statistical test to check whether the improvements are statistically significant.
> > >
> > >in the revised version of the paper, we will run the experiments with different splits to show the reproducibility of the results and statistical >significance.
> >
> > I have not seen any change in the results in Table 2 on the datasets that were used in the original paper. I thus assume that results in the new Table 2 are still obtained with a single train/test split (but I understand it takes time to run such experiments). A statistical test has been added, but I'm not sure how it is actually carried out. To me, a Wilcoxon signed-rand test is a way to compare two algorithms across several datasets. Using this test, it's thus not possible to make a statement about whether the difference between two algorithms is significant on a given dataset. In the light of this, I don't understand the statement "the improvements are significant according to Wilcoxon signed-rank test with p − value ≪ 0.001, except for datasets Click and Kick" in the paper. Why can you say "except for datasets Click and Lick"?
> >
> > > In fact, we set learning rate in such a way that the cross validated optimal point is close to the 2500-th iteration to ensure convergence of the
> > > training proceess. B=5000 was previously shown enough for achieving SOTA result. We agree that other settings are also interesting (for
> > > example, with a bounded size of the model) Thank you for all suggestions, we consider providing additional empirical evidences!
> >
> > It's not clear that this is appropriate (I'm not sure what "to ensure convergence of the training process" actually means). It seems to me that you have then tuned the learning rate so that the error curve goes through some minimum, which is a way to make pruning more relevant and thus introduce a bias in favour of your approach. What happens if you tune instead the learning rate to obtain the best cross-validated performance at B=5000? Is pruning still useful then?

---

> > > ### Author Response · Authors · 2021-11-30
> > > **Answer to comments**
> > >
> > > We apply the pairwise Wilcoxon test to the test sets’ results obtained from the default and pruned models. Namely, the test set was divided into parts, and the qualities of two models on each part form one observation pair. Then we apply the pairwise Wilcoxon test to the obtained set of pairs.
> > >
> > > The method of selecting a global optimal number of iterations we use is conventional. In practice, the model is trained for a sufficiently large number of iterations so that the overfitting point is reached approximately in the middle of the learning curve to ensure convergence and to deal with possible double descent. Then the number of iterations is set to that optima. The same situation would occur if we were targeting the 5,000th iteration. In this case, we would train our model for approximately 10,000 iterations to ensure that we do not miss any better point larger than 5,000. The idea of our pruning is still relevant since we select an optimal stop for each region adaptively instead of setting the global parameter to 5,000.

---

### Decision · Program_Chairs · 2022-01-20

**Decision:**

Reject

**Comment:**

While the reviewers agree that the paper contains interesting ideas and the method is elegant, it unfortunately does not meet the bar for acceptance. I strongly encourage the authors to revise their paper, in particular using the numerous comments made throughout the discussion phase; for example:

* It is important that the authors polish their work, in particular for the updates provided (e.g. Figure 3, see EFwa)

* Reviewers pointed the lack of updates on important claims by the authors (in particular the claim regarding clustering vs decision trees, see EFwa, the comments on the lack of diverse datasets, see meXP, )

* Some answers might have gained in clarity, such as the reply to EFwa on the application and conclusions following Wilcoxon sign test.